

# Prediction of COVID-19 epidemic situation via fine-tuned IndRNN

Zhonghua Hong[1,2], Ziyang Fan[1], Xiaohua Tong[2], Ruyan Zhou[1], Haiyan Pan[1], Yun Zhang[1], Yanling Han[1], Jing Wang[1], Shuhu Yang[1], Hong Wu[1] and Jiahao Li[1]

[1] College of Information Technology, Shanghai Ocean University, Shanghai, China
[2] College of Surveying and Geo-Informatics, Tongji University, Shanghai, China

Corresponding authors
Zhonghua Hong,
zhhong@shou.edu.cn
Xiaohua Tong, xhtong@tongji.edu.cn

## ABSTRACT

The COVID-19 pandemic is the most serious catastrophe since the Second World War. To predict the epidemic more accurately under the influence of policies, a framework based on Independently Recurrent Neural Network (IndRNN) with fine-tuning are proposed for predict the epidemic development trend of confirmed cases and deaths in the United Stated, India, Brazil, France, Russia, China, and the world to late May, 2021. The proposed framework consists of four main steps: data pre-processing, model pre-training and weight saving, the weight fine-tuning, trend predicting and validating. It is concluded that the proposed framework based on IndRNN and fine-tuning with high speed and low complexity, has great fitting and prediction performance. The applied fine-tuning strategy can effectively reduce the error by up to 20.94% and time cost. For most of the countries, the MAPEs of fine-tuned IndRNN model were less than 1.2%, the minimum MAPE and RMSE were 0.05%, and 1.17, respectively, by using Chinese deaths, during the testing phase. According to the prediction and validation results, the MAPEs of the proposed framework were less than 6.2% in most cases, and it generated lowest MAPE and RMSE values of 0.05% and 2.14, respectively, for deaths in China. Moreover, Policies that play an important role in the development of COVID-19 have been summarized. Timely and appropriate measures can greatly reduce the spread of COVID-19; untimely and inappropriate government policies, lax regulations, and insufficient public cooperation are the reasons for the aggravation of the epidemic situations. The code is available at https://github.com/zhhongsh/COVID19-Precdiction. And the prediction by IndRNN model with fine-tuning are now available online (http://47.117.160.245:8088/IndRNNPredict).

## INTRODUCTION

The coronavirus disease 2019 (COVID-19) is brought on by infection from Severe Acute Respiratory Syndrome (SARS) Coronavirus 2, and the reports of related cases were first released by Wuhan, Hubei Province, China in December 2019 (*Zhou et al., 2020*). The widespread COVID-19 epidemic is a serious threat, and has become one of the most challenging global catastrophes facing mankind since the Second World War (*China Daily, 2020*). On March 11th, 2020, the global development of COVID-19 was assessed by

the World Health Organization (WHO) as having met the characteristics of a pandemic (*World Health Organization, 2020a*). The mortality rate of COVID-19 is estimated to be between 2% and 5%, *i.e.,* lower than that of SARS and Middle East Respirator Syndrome (*Gasmi et al., 2020*; *Wu, Chen & Chan, 2020*). However, COVID-19 has a higher infection rate than bat-like SARS, and its pathogenicity is between that of SARS and bat-like SARS (*Benvecnuto et al., 2020*). The heart may be damaged and develop myocardial inflammation after recovery from COVID-19 (*Puntmann et al., 2020*). In addition, the widespread nature of the disease also slows down activity on a national and global level, along with aggravating unemployment and hunger (*Mckibbin & Fernando, 2020*; *Armario, 2020*).

On January 15th, 2020, the Chinese Center for Disease Control and Prevention (China CDC) initiated a first-level emergency response (*Li et al., 2020a*). A series of policies began to be implemented since January 23th, 2020 (*Yang et al., 2020*), and the epidemic had stabilized by March 2020. In contrast, the epidemic situation overseas are not very optimistic. The number of infected people continues to maintain substantial increase of tens of thousands. According to the report by the WHO on April 25th, 2020, the global epidemic has infected more than 146 million people, with 3,092,497 deaths, and the epidemic in the Americas, Europe, and South-East Asia are the worst, especially in India and the United States (*World Health Organization, 2020b*).

Predicting the development trend based on the increase of cases is useful for the adjustment of epidemic prevention policy. However, in the current epidemic prediction work, the models used are complex and slow (*Yang et al., 2020*; *Bandyopadhyay & Dutta, 2020*), and some methods are fast but not effective (*Huang et al., 2020a*). In addition, the number of cumulative increases is not stable but variable, particularly, a sudden rapid or modest increase affects the stability of modelling and therefore the accuracy of predicting future trends. Thus, it is necessary to propose a model with coexistence of high precision, high speed, and low model complexity to predict and analyze the development tendency of COVID-19 with more efficiency and accuracy, and to summarise the meaningful and positive policies.

In this paper, a framework combining IndRNN model and fine-tuning strategy is proposed. Different from the existing models in the prediction of COVID-19, the fine-tuning strategy is added to reduce error and time. The proposed framework consists of four steps. First, the original data are pre-processed, including normalization and sequence data generation. Then, the IndRNN model is used to iteratively train to learn information or features hidden in COVID-19 sequence data, and weight parameters is obtained. Third, the fine-tuning strategy is added to load the recently updated data into the model with existing weight parameters for partial parameter adjustment. Finally, the model that has learned the characteristics or information is used to predict the trend, and the prediction results are compared with the real data for verification, and the policy events and social economic influences under the epidemic situation are analyzed. The contributions in this paper as follows:

(1) A framework based on IndRNN and fine-tuning strategy, which can be adept at longer sequence data, was used in COVID-19 epidemic prediction with more accuracy and high speed. IndRNN is used to retain effective information and better learn the characteristics

of changeable sequence data in COVID-19 due to the advantage of independent neurons prevents gradient explosion and disappearance phenomenon. The fine-tuning strategy is utilized to further improves the accuracy in a short time, and it avoids the consumption brought by retraining after data update.

(2) The fitting performance of the proposed framework based on IndRNN and fine-tuning are verified among the cumulative cases of India by comparing with the Long-Short-Term-Memory (LSTM), bidirectional LSTM (Bi-LSTM), Gated-Recurrent-Unit (GRU), Stacked_Bi_GRU, Convolutional Neural Network_LSTM (CNN_LSTM), Deep_CNN models. And the prediction accuracy of the model is validated by the error between the prediction results and the real data of the next four weeks in the cumulative case data of the United State, India, Brazil, France, Russia, China, and world.

(3) The growth of the cumulative population is combined to analyze the positivity of epidemic policies and activities. The phenomenon is found that the same proactive policies have been implemented inconsistently in different countries because of lax regulation and inadequate public cooperation.

The rest of the paper is organized as follows: 'Related work' describes the related work of this research direction. 'Materials & Methods' introduces the data sources and study area, the proposed framework, and performance metrics; the experimental results are presented in 'Results'. In 'Discussion', impact of fine-tuning on the prediction and validation accuracy, and the influence of policies are discussed. Finally, some conclusions are drawn in 'Conclusions'.

## RELATED WORK

There are two main methods for forecasting the epidemic development of COVID-19: mathematical model and deep learning model.

A typical mathematical model in epidemic dynamics is the Susceptible-Exposed-Infectious-Removed (SEIR) model, using mathematical formulas to reflect the relationships between the flows of people at four states: susceptible, exposed, infectious, and recovered (*Fang, Nie & Penny, 2020*). The SEIR model was used to effectively predict the peaks and sizes of COVID-19 epidemiological data with sufficient fitting performance (*Yang et al., 2020*; *Fang, Nie & Penny, 2020*). A modified SEIR model also showed a good effect for predicting the peaks and sizes (*Yang et al., 2020*). The peak deviation of the another modified SEIR model in predicting epidemiological data in China was 3.02% (*Fang, Nie & Penny, 2020*). However, SEIR focuses on predicting trends in sensitive, exposed, infected, and recovered groups (*Yang et al., 2020*; *Fang, Nie & Penny, 2020*), rather than cumulative confirmed and death cases. Moreover, it is necessary to comprehensively consider the changes in some parameter values as affected by the changes in epidemic policies and regional differences, such as the effective reproductive number, number of contacts, and infection rate. And these parameters are not easy to obtain, and are uncertain.

Deep learning automatically extracts the features from the data and builds the model without the need for other specific parameters. This method generates a series of sequence data from COVID-19 epidemiological data and looks for regular changes in the sequence

data. Some deep learning models based on LSTM was used to predict COVID-19 trend, and the results demonstrated that LSTM has good prospects for predicting the trend; however, the fitting effect needs to be improved (*Yang et al., 2020*; *Zandavi, Rashidi & Vafaee, 2020*), too many internal parameters of LSTM increase the complexity of model training, and it can only learn forward. The Bi-LSTM, which adds bidirectional learning capability on LSTM, and the GRU, which makes gates simplification on the internal structure of LSTM, are used by *Shahid, Zameer & Muneeb (2020)* for epidemic prediction. Moreover, a stacked-Bi-GRU model was applied for COVID-19 trend forecast, owing to the learning adequacy of the bidirectional cyclic network (*Bandyopadhyay & Dutta, 2020*). LSTM and GRU solve the problem of gradient vanishing explosion of RNN, but there is a gradient attenuation problem of layer. In addition, the entanglement of neurons in the same layer makes their behavior difficult to interpret.

The existence of these multiple gate operations in the recurrent unit of the RNN variants, which is LSTM, Bi-LSTM, GRU, makes the parameters complicated. CNN does not need many parameters in virtue of the advantage of weight sharing, in comparison with these recurrent networks. A deep CNN model (*Huang et al., 2020b*) and a CNN-LSTM model (*Dutta, Bandyopadhyay & Kim, 2020*) were proposed for analysing and predicting the number of confirmed cases in China. However, the training speed of deep CNN is fast, but the effect is not significantly improved. The CNN_LSTM model combines CNN and LSTM, but increases the model complexity.

In the above deep learning network model, high precision, high speed, and low model complexity cannot coexist. And the cumulative epidemic data is time series, so recurrent networks are more suitable for processing such serialized data in comparing with CNN. Similar to LSTM, GRU, Bi-LSTM, IndRNN is one of the variants of RNN, which can learn longer sequence data, and it has no redundant gate operations, fewer parameters, and can be more easily trained (*Li et al., 2018*). Simultaneously, IndRNN is designed to solve the problems of gradient disappearance and explosion, and can be utilized to process sequence data like LSTM, GRU and Bi-LSTM. IndRNN was used to learn relationships between plant gene sequences, overcoming the uncertainty of artificially acquired traits, and has higher accuracy than LSTM (*Zhang et al., 2020*). Therefore, Indrnn is adopted as the basic model in this paper to improve the accuracy of epidemic prediction.

However, the epidemic data are updated daily, and the network weights need to be retrained after obtaining the data of the last few weeks, which are time-consuming. Fine-tuning can transfer the trained network model parameters to the required network for partial parameter adjustments, without the need to train from scratch. *Tajbakhsh et al. (2016)* used a fine-tuned pre-trained CNN network for medical image analysis and found that the effect was better than training all data from scratch CNN. (*Boyd, Czajka & Bowyer, 2020*) demonstrated that fine-tuning existing network weights to extract iris features was more accurate. Therefore, a fine-tuning strategy is added in this study to further improve the accuracy and speed of prediction.

Combined with the above, in order to achieve rapid and more accurate epidemic prediction, a framework based on IndRNN and fine-tuning to predict COVID-19 epidemiological data is proposed in this paper.

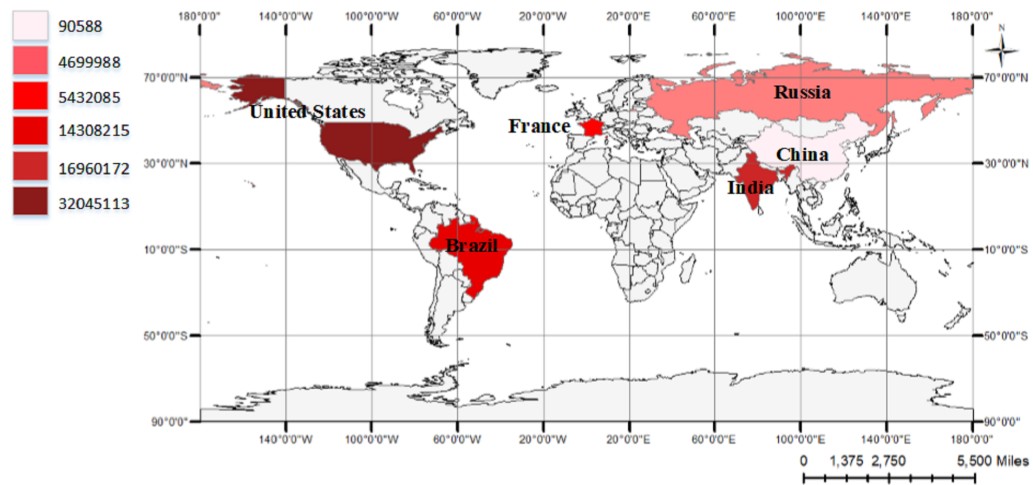

**Figure 1** **China and top five countries of confirmed COVID-19 cases worldwide.** The data on the left represents the amount of cumulative confirmed cases in China and five countries with severe epidemic situation as of April 24, 2021.

## MATERIALS & METHODS

### Data sources and study area

The study area of this research is shown in Fig. 1. In total, six countries are involved, which are China, the United States, India, Brazil, France and Russia. China is the country where the first COVID-19 has been reported in the world. The epidemiological data of COVID-19 in China between January 18th, 2020 and May 22th, 2021 were downloaded from http://www.nhc.gov.cn/xcs/yqtb/list_gzbd.shtml (*National Health Commission of the People's Republic of China, 2020*). The United States, India, Brazil, France and Russia are the most severely affected countries in the world during this period. The data of these top five countries were download from a dataset based on global COVID-19 epidemic statistics, as published by https://github.com/CSSEGISandData/COVID-19 (*Johns Hopkins University in the United States, 2020*). In addition, the cumulative overseas epidemic data also analyzed, the data were retrieved from https://voice.baidu.com/act/newpneumonia/newpneumonia/?from=osari_pc_3 (*Baidu, 2020*). The Chinese and overseas COVID-19 data were combined to generate the numbers of cumulative confirmed cases and cumulative deaths worldwide. It should be noted that all of the three websites share the same definition of cumulative confirmed and death cases, which ensures the consistency of the data. The detailed information of the data used in this study is presented in Table 1.

### Methods

The flowchart of the proposed framework is illustrated in Fig. 2. It mainly consists of four steps as follows: (1) the cumulative confirmed and death cases obtained from various websites are first preprocessed to generate sequence data of size in a uniform range, and the sequence data are divided into training data, fine-tuning data, and testing data; (2) the deep learning model trains and learns the features using the training data, then the weights

**Table 1  Detailed information of the COVID-19 epidemiological data used in this study.**

| Country /Region | Statistical period | | Source | |
| --- | --- | --- | --- | --- |
| | Confirmed | Deaths | Institution | Website |
| China | 2020/01/18~ 2021/5/22 | 2020/01/18~ 2021/5/22 | National Health Commission of the People's Republic of China | http://www.nhc.gov.cn/xcs/yqtb/list_gzbd.shtml |
| United States | 2020/01/25~ 2021/5/22 | 2020/02/29~ 2021/5/22 | | |
| India | 2020/02/01~ 2021/5/22 | 2020/03/14~ 2021/5/22 | | |
| Brazil | 2020/02/29~ 2021/5/22 | 2020/03/21~ 2021/5/22 | | |
| France | 2020/03/07~ 2021/5/22 | 2020/02/15~ 2021/5/22 | Johns Hopkins University in the United States | https://github.com/CSSEGISandData/COVID-19 |
| Russia | 2020/02/01~ 2021/5/22 | 2020/03/21~ 2021/5/22 | | |
| Global | 2020/01/15~ 2021/5/22 | 2020/02/15~ 2021/5/22 | National Health Commission of the People's Republic of China; Baidu | http://www.nhc.gov.cn/xcs/yqtb/list_gzbd.shtml |

are obtained. The training process is the process of establishing the optimal relationship between the training data and the training labels, in which the training data is the input and the training label is the perfect output. Through continuous iterative training, the characteristics of training data are found, and the parameters in the network model are adjusted to continuously fit the actual output and training labels; (3) the fine-tuning data is loaded with the weight file, and the parameters are adjusted by fine-tuning model, continuously reducing the error between the fine-tuning data and the fine-tuning labels; (4) though applying the model built from the previous training and fine-tuning, the testing data are used for testing, later trends are predicted, and validated using true data. The testing data are input for testing by applying the model with weights, to determine the gap between the actual output of test data and the test labels. After that, the model is used for direct trend prediction, and the prediction results are compared with the real updated data to judge the prediction performance of the proposed framework.

Specially, seven deep learning models which are LSTM, Bi-LSTM, GRU, Stacked_Bi_GRU, CNN_LSTM, Deep_CNN, and IndRNN are used in this research for model comparison. As shown in Fig. 3, the structures of the LSTM, Bi-LSTM, GRU, and IndRNN models are approximately the same. The flatten layer in LSTM, Bi-LSTM and GRU models is used to convert three-dimensional data into two-dimensional data. The IndRNN model does not add the Flatten layer because the output of the IndRNN layer is 2-dimensional data. The details of the proposed framework are described below.

## Pre-processing

Pre-processing includes MinMaxScaler operation and sequence data generation. To keep the data at the same order of magnitude and facilitate characteristic analysis and model convergence, the cumulative confirmed cases and deaths data are scaled to 0~1 after

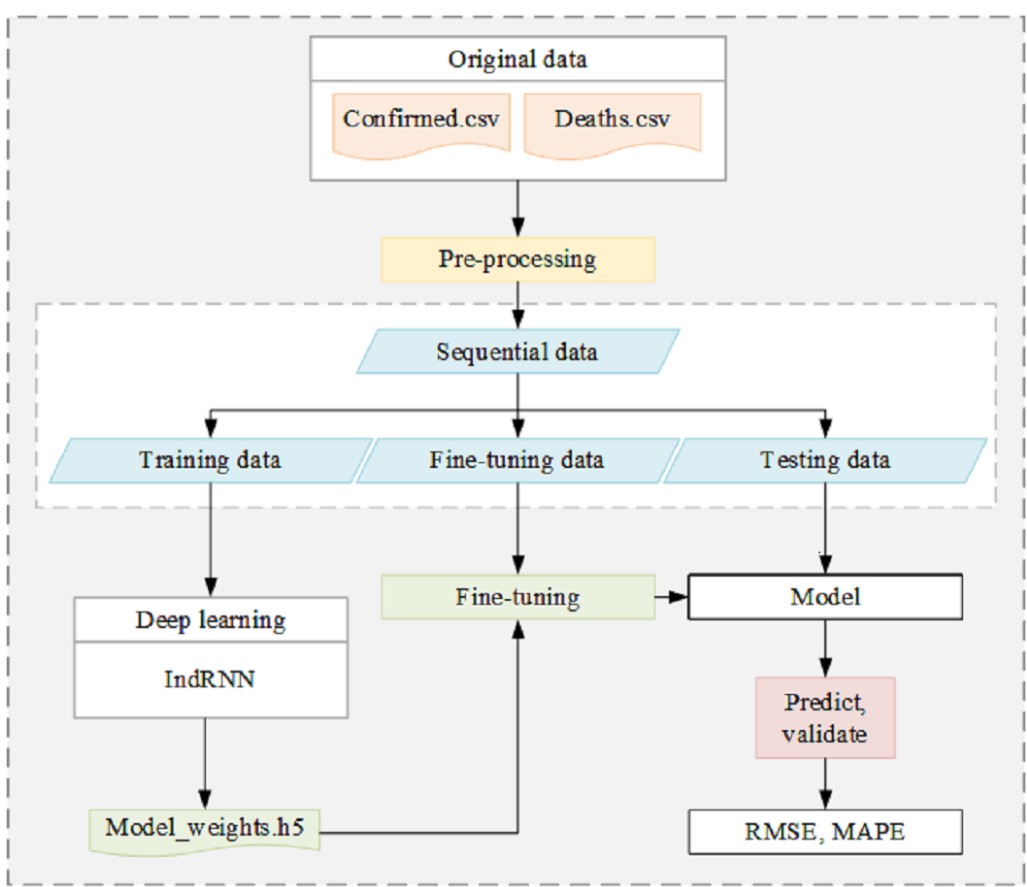

**Figure 2** Flow chart of the proposed scheme.

MinMaxScaler operation based on the minimum and maximum values, and the original proportion of the data is retained.

Sequence data generation is a crucial step, and it is the premise of sequence model training. The sequence data can reflect changes in which no regularity can be traced in single data. Therefore, the individual data were organised into sequential data. As shown in Fig. 4, the dimensionality of the individual data for $n$ weeks was $(n, 1)$, a window with a sequence length of $x$ and step size of 1 was initially selected for sliding orderly through the weekly data. All of the data contained in the window was considered as a data sequence, and the next week data outside the window was the corresponding label data. After the window slid through all of the weekly data, the sequence data with dimension of $(n-x, x)$ was generated. It is noteworthy that the content of sequence data represents features. For example, when the sequence length is 3, the data from the first to the third form a sequence, and the data from the fourth day is the label data; next, the data of days 2 to 4 form a sequence, and the data of day 5 is label data. The sequence data generation is completed until the label data reaches the last data of the input data.

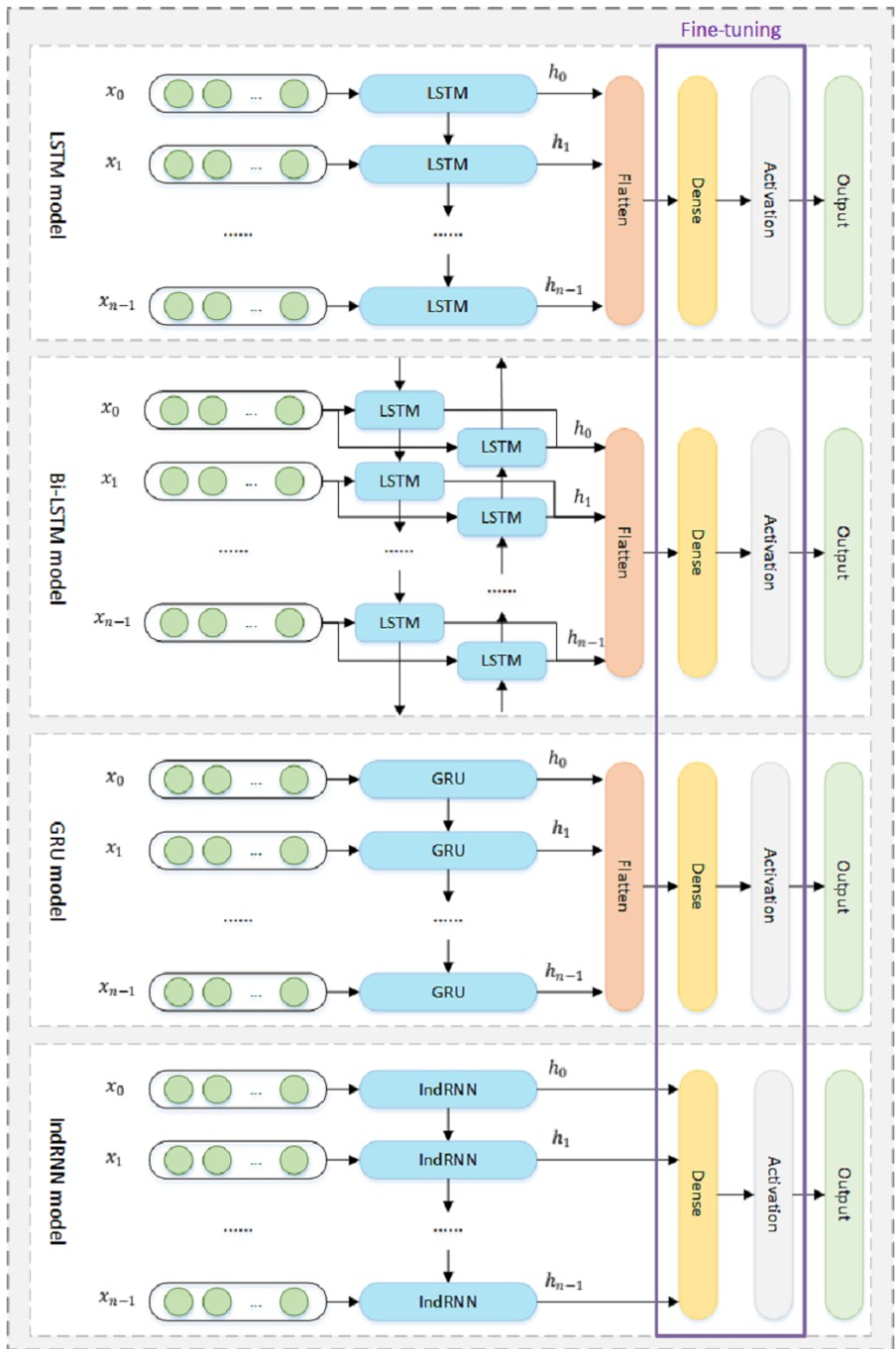

**Figure 3** **The structures of LSTM, Bi-LSTM, GRU and the IndRNN model.** The purple box indicates the network layers that need to be fine-tuned.

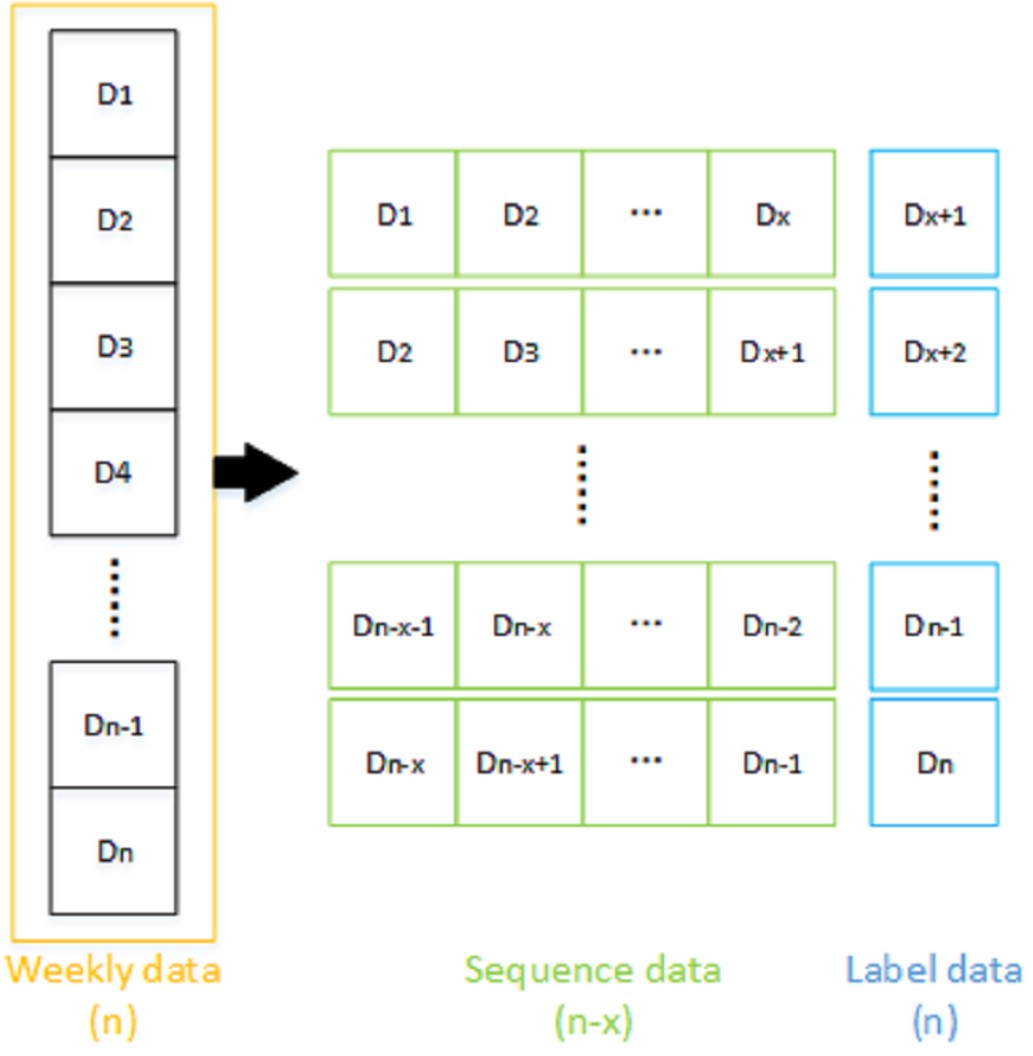

**Figure 4** Sequence data generation.

## LSTM and Bi-LSTM

In RNN, too many cyclic units may lead to the loss of previously learned rule information and the existence of long-term dependence, and can lead to the problems such gradient disappearance or gradient explosion (*Goodfellow, Bengio & Courville, 2016*). LSTM is mainly used to learn and overcome the long-term dependence of RNN (*Hochreiter & Schmidhuber, 1997*). The internal units of the LSTM include the memory cell, forget gate, input gate, and output gate (*Goodfellow, Bengio & Courville, 2016*). In the LSTM internal unit, the memory cell carries the necessary information transfers between the LSTM internal circulation units, thereby solving the problem of gradient disappearance, and learning long-term dependence (*Olah, 2015*). The sigmoid layers in the three gates constrain the value between 0 and 1, so as to determine which information should be saved or forgotten (*Goodfellow, Bengio & Courville, 2016*; *Olah, 2015*). When the information

flow enters the circulation unit, the operation is as follows (*Olah, 2015*):

$$forget_t = \sigma\left(W_{forget} \cdot [h_{t-1}, x_t] + b_{forget}\right) \tag{1}$$

$$input_t = \sigma\left(W_{input} \cdot [h_{t-1}, x_t] + b_{input}\right) \tag{2}$$

$$\tilde{C}_t = \tanh\left(W_C \cdot [h_{t-1}, x_t] + b_C\right) \tag{3}$$

$$C_t = forget_t * C_{t-1} + input_t * \tilde{C}_t \tag{4}$$

$$output_t = \sigma\left(W_{output} \cdot [h_{t-1}, x_t] + b_{output}\right) \tag{5}$$

$$h_t = output_t * \tanh(C_t) \tag{6}$$

In the above, $t$ stands for the moment, $forget_t$, $input_t$, and $output_t$ represent the output after the activation function $\sigma$ in the forget gate, input gate and output gate, respectively. $W_{forget}$, $W_{input}$, $W_{output}$, $W_C$, $b_{forget}$, $b_{input}$, $b_{output}$, $b_C$ symbolize the weight and bias offset of the three gates and memory cell, respectively. $h_{t-1}$ is the output at the previous moment, $x_t$ is the input at the current moment. The updated memory cell $C_t$ is obtained by adding $forget_t * C_{t-1}$ which represents unnecessary information to be discarded, and $input_t * \tilde{C}_t$ which represents the information to be updated. Finally, the information in the memory unit $C_t$ is controlled by the output gate to return the final output $h_t$ for this LSTM cell.

LSTM can only predict the output based on the previous content, but the later information will also help understand the current text. Therefore, the Bi-LSTM proposed by *Schuster & Paliwal (1997)* performs bidirectional input and makes full use of the context information. The forward input is the sequence input at time $t$ and the output at time $t-1$, and the backward input is the sequence input at time t and the output at time $t+1$. The final output is a combination of forward output $h_{forword} = (hf_0, hf_1, \ldots, hf_{n-1})$ and reverse output $h_{backword} = (hb_{n-1}), hb_{n-2}, \ldots, hb_0)$.

## GRU

The difference between GRU and LSTM is that the GRU changes the forgetting gate and input gate in LSTM to update gate, and reduces the amount of gate controls (*Chung et al., 2014*). The GRU transmits information directly through the hidden state, rather than through the memory cell. The input information of the current moment and the output information of the previous moment are firstly determined by the update gate whether to be updated. The reset gate is then used to control whether the information is set to 0, *i.e.,* to determine the amount of information to be retained in the candidate information. Finally, it is up to update gate to control how much information in the output at the previous

time is forgotten and how much information is added to the hidden information, and the retained information forms the output of the GRU recurrent unit at the current moment. The main formulas of GRU internal structure are as follows (*Chung et al., 2014*):

$$update_t = \sigma\left(W_{update} \cdot \left[h_{t-1}, x_t\right]\right) \tag{7}$$

$$reset_t = \sigma(W_{reset} \cdot \left[h_{t-1}, x_t\right]) \tag{8}$$

$$\tilde{h}_t = \tanh\left(W \cdot \left[reset_t * h_{t-1}, x_t\right]\right) \tag{9}$$

$$h_t = \left(1 - update_t\right) * h_{t-1} + update_t * \tilde{h}_t \tag{10}$$

Where, $update_t$, $reset_t$ represent the output after the activation function $\sigma$ in the update gate and reset gate, respectively. $\tilde{h}_t$ is candidate information, and $h_t$ represents the output at the current moment $t$.

## IndRNN

IndRNN solves the problem of neuron independence in RNN and the gradient attenuation that appears in LSTM and GRU, and can handle longer sequences. In addition, gate computation is added to LSTM and GRU, which increases the computational complexity of the network layer. In IndRNN, there are few parameters, and the neuron only receives input from this moment and its own state input from the previous moment at any given moment, making each neuron independent (*Li et al., 2018*). The state update formula of the hidden layer is (*Li et al., 2018*):

$$h_t = \sigma(Wx_t + u \odot h_{t-1} + b) \tag{11}$$

Here, $W$ and $u$ are the weights of a certain neuron at the current and previous moment, respectively.

## Fine-tuning

Fine-tuning refers to copying the weight of the trained network model to the network that needs to be used, and continuing to train and adjust part of the weight (*Tajbakhsh et al., 2016*). The advantage of fine-tuning is that it can achieve better results in a shorter time than training the network from scratch, owing to pre-training using a large amount of data (*Tajbakhsh et al., 2016*). The fine-tuning model in this paper is consistent with the pre-training model, as shown in Fig. 3. Firstly, the fine-tuning model copies and transfers the weight of the pre-training model to itself. Then, the parameters of the net layer before the dense layer in the Stacked_Bi_GRU, CNN_LSTM, Deep_CNN models, and the LSTM layer, Bi-LSTM layer, GRU layer, IndRNN layer, are frozen, respectively, and do not participate in the following training. The parameters of the dense layer and the activation layer of each fine-tuning model are in an active state, waiting for adjustment. Finally, import fine-tuning data to start fine-tuning training. The error between feature sequence data and label data is gradually narrowed by iterative training with the full utilization of the features or knowledge acquired by the pre-training model.

## Assessment metrics

To evaluate the effectiveness of the model, the RMSE and MAPE were used to access the fitting performance between the output (prediction data) and label data of each sequence data. The quations are as follows:

$$RMSE(L,h) = \sqrt{\frac{1}{m}\sum_{i=1}^{m}(L_i - h_i)^2} \qquad (12)$$

$$MAPE(L,h) = \frac{100}{m}\sum_{i=1}^{m}\left|\frac{L_i - h_i}{L_i}\right| \qquad (13)$$

Here, $m$ denotes the number of sequential data, and $h_i$ and $L_i$ represent the output of each sequence $i$ after testing in model and the corresponding label data, respectively. The smaller the value of RMSE and MAPE, the better the prediction effect, which means that the error between prediction data and label data is smaller.

# RESULTS

This paper aims to achieve prediction of epidemic trends with more accurate by proposed framework based on IndRNN and fine-tuning strategy on COVID-19 epidemiology data. The deadline for training data is March 13th, 2021, the data from 3/20/2021 to 4/24/2021 is used for fine-tuning and testing, and we have taken cases from 5/1/2021 to 5/22/2021 for validation. The deep learning environment of our experiment was mainly built based on the Ubuntu 16.04 system environment, which encoded by python Keras with the support of Intel Core i9-10920X CPU @3.50 GHz ×24.

Three tasks are completed in this paper: (1) model comparison by LSTM, Bi-LSTM, GRU, Stacked_Bi_GRU, CNN_LSTM, Deep_CNN, and IndRNN models, (2) the fine-tuned IndRNN model is utilized to predict the number of confirmed cases in the United Sates, India, Brazil, France, Russia, China and the world, and verify their accuracy, and (3) the growth status of the cumulative cases in 6 countries in combination with current policies are analyzed. All results in this section use the values in Table 2. The experimental procedure for each model follows the steps in the method. Moreover, during fine-tuning, the numbers of layers in the seven models that do not participate in training are set as shown in Table 3.

## Comparison of models

The performance of seven different models which are LSTM, Bi-LSTM, GRU, Stacked_Bi_GRU, CNN_LSTM, Deep_CNN, and IndRNN models are compared based on the COVID-19 statistical data of India. The reason for this selection lays on the fact that India is the worst-affected country under the current situation, the weekly increase of cases is dramatically higher than other countries, with vast changing data. Before adding the updated data, the six countries and the world used in this paper have at least 55 weeks of data. To find the most suitable sequence length, and to ensure that the number of

**Table 2  Consistent parameters and their values in LSTM, Bi-LSTM, GRU, Stacked_Bi_GRU, CNN_LSTM, Deep_CNN, and IndRNN models.**

| Parameters | Values |
| --- | --- |
| Loss | Mean squared error |
| Optimizer | Adam |
| Batch size | 1 |
| Epoch in training | 3,000 |
| Epoch in testing | 3,000 |
| Sequence length | 45 weeks |

**Table 3  The numbers of network layers of the model and frozen network layers when fine-tuning.**

| Model | Quantity of total layer | Quantity of frozen layer when fine-tuning |
| --- | --- | --- |
| LSTM | 5 | 3 |
| Bi-LSTM | 5 | 3 |
| GRU | 5 | 3 |
| IndRNN | 4 | 2 |
| Stacked_Bi_GRU | 14 | 10 |
| CNN_LSTM | 19 | 14 |
| Deep_CNN | 8 | 7 |

training data is more than fine-tuning data, Indian dataset was used for training by LSTM, Bi-LSTM, GRU, Stacked_Bi_GRU, CNN_LSTM, Deep_CNN, and IndRNN models with sequence length between 5 and 45 weeks (separated by 4 weeks, *i.e.,* one month) as single data cannot become sequence data. The best results were obtained when the sequence length was 45.

After normalization and sequence data generation, the dimensions of the sequence data generated by using weekly cumulative confirmed and death cases in India are (20, 1, 45) and (14, 1, 45), and the corresponding label data are 20 and 14 respectively. The last 6 of them are used for fine-tuning and testing, and the other data are used for training.

These 7 models use Indian COVID-19 data to save the weights after pre-training, respectively. Under the condition of freezing the network layer before the fully connected layer, the pre-trained models load the corresponding weights for iterative fine-tuning by the fine-tuning data. In this process, to find the appropriate amount of fine-tuning data, 1, 3, and 5 pieces of fine-tuning data are used for fine-tuning, and the number of corresponding testing data are 5, 3, and 1 respectively.

Figure 5, Tables 4, and 5 show the comparison results. Where, in Table 4, "f" and "t" in the column "Split" represent the number of fine-tuning data and testing data, respectively. As can be seen from the tables and the figure, IndRNN model shows the best performance for both with no-fine-tuning and fine-tuning models, compared to LSTM, Bi-LSTM, GRU, Stacked_Bi_GRU, CNN_LSTM, and Deep_CNN models. Where, IndRNN model had the least MAPE and RMSE before and after fine-tuning, the lowest total runtime when using cumulative diagnosis data, the second lowest total runtime when using cumulative death data, and minimum number of total parameters. The MAPE and RMSE

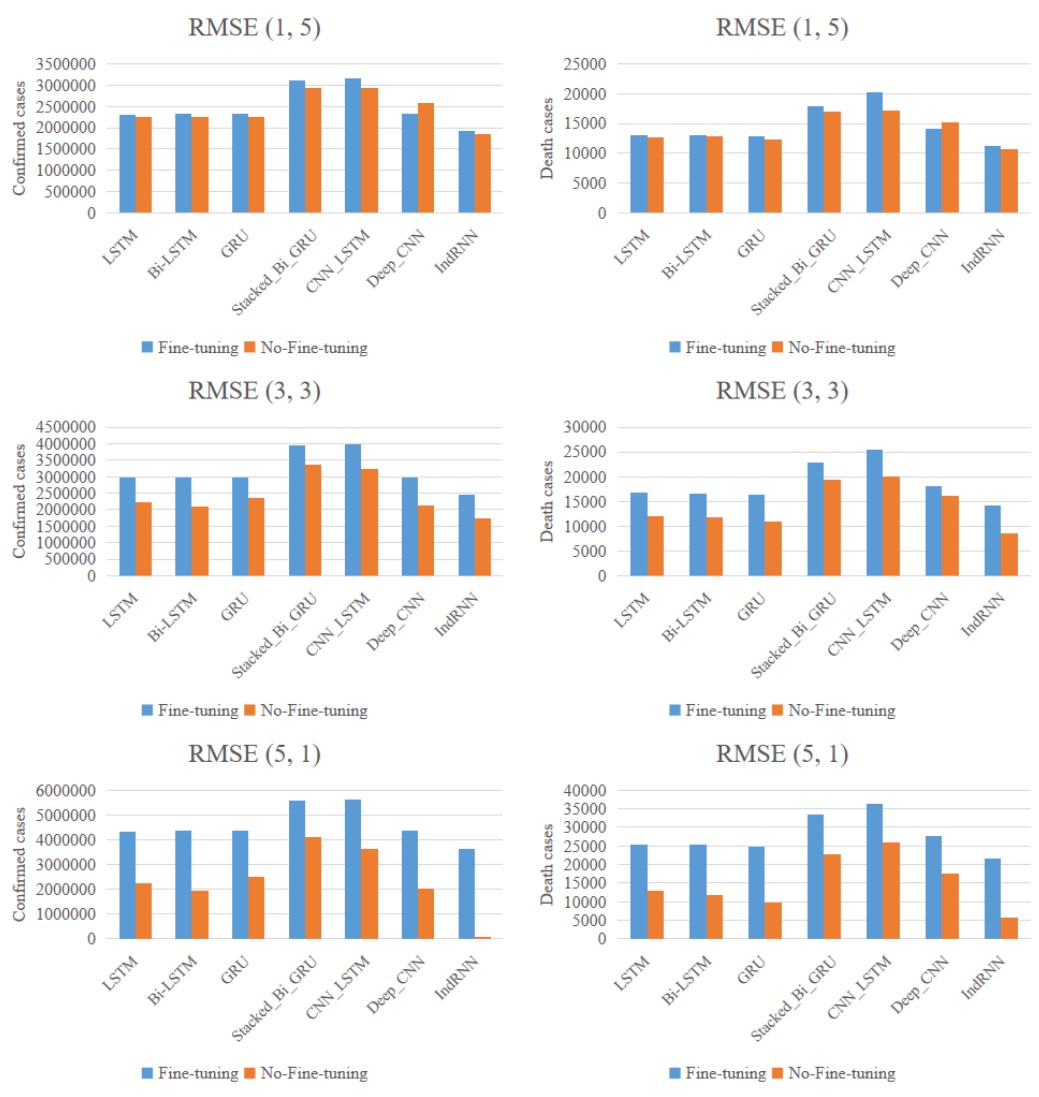

**Figure 5** **Comparison among LSTM, Bi-LSTM, GRU, IndRNN in terms of RMSE.** Prediction of cumulative confirmed COVID-19 cases in India. The sequence length of 45 weeks is used in the model.

of LSTM, BI-LLSTM and GRU are similar, and the effect ranking of the remaining models is Deep_CNN, Stacked_Bi_GRU, CNN_LSTM, respectively. After fine-tuning, the RMSE and MAPE of the seven models are all significantly decreased, especially for IndRNN, the MAPE of which is decreased by 0.27%∼12.36%, 0.12%∼14.29%, 0.3%∼10.91%, 0.69%∼8.89%, 2.21%∼11.84%, −1.59%∼13.79%, and 0.36%∼20.94%, respectively. The best test results were obtained when five fine-tuning data were used to fine-tune the parameters of the pre-trained IndRNN model. It is proved that fine-tuning plays a very important role in reducing errors. In addition, IndRNN also exhibits superiority in computational efficiency. That is also true for that with fine-tuning. Figure 6 shows the fitting performance of the IndRNN model. The predicted value is very close to the true value and the two lines are

**Table 4  Comparison among LSTM, Bi-LSTM, GRU, Stacked_Bi_GRU, CNN_LSTM, Deep_CNN, and IndRNN in terms of MAPE.** Prediction of cumulative confirmed COVID-19 cases in India.

| Data | Model | Training time(s) | Split (f, t) | MAPE (%) | | Fine-tuning time(s) |
|---|---|---|---|---|---|---|
| | | | | No-fine-tuning | Fine-tuning | |
| Confirmed cases | LSTM | 733.38 | 1, 5 | 11.69 | 11.03 | 21.40 |
| | | | 3, 3 | 17.13 | 12.07 | 47.19 |
| | | | 5, 1 | 25.60 | 13.24 | 77.71 |
| | Bi-LSTM | 1296.12 | 1, 5 | 11.72 | 11.10 | 28.82 |
| | | | 3, 3 | 17.19 | 11.10 | 68.27 |
| | | | 5, 1 | 25.72 | 11.43 | 105.57 |
| | GRU | 581.84 | 1, 5 | 11.72 | 11.05 | 20.14 |
| | | | 3, 3 | 17.18 | 12.86 | 45.32 |
| | | | 5, 1 | 25.70 | 14.79 | 70.85 |
| | Stacked_Bi_GRU | 5291.32 | 1, 5 | 17.02 | 15.33 | 69.52 |
| | | | 3, 3 | 23.70 | 19.22 | 183.36 |
| | | | 5, 1 | 33.01 | 24.12 | 287.70 |
| | CNN_LSTM | 3006.03 | 1, 5 | 17.58 | 15.29 | 41.15 |
| | | | 3, 3 | 23.88 | 18.45 | 104.69 |
| | | | 5, 1 | 33.16 | 21.32 | 170.93 |
| | Deep_CNN | 261.37 | 1, 5 | 11.63 | 13.22 | 13.11 |
| | | | 3, 3 | 17.21 | 11.41 | 29.44 |
| | | | 5, 1 | 25.84 | 12.05 | 45.71 |
| | IndRNN | 248.72 | 1, 5 | 9.74 | 9.11 | 14.85 |
| | | | 3, 3 | 14.17 | 8.85 | 34.96 |
| | | | 5, 1 | 21.40 | 0.46 | 57.08 |
| Confirmed deaths | LSTM | 413.85 | 1, 5 | 5.29 | 5.02 | 20.93 |
| | | | 3, 3 | 7.94 | 5.19 | 48.49 |
| | | | 5, 1 | 13.21 | 6.77 | 74.72 |
| | Bi-LSTM | 747.53 | 1, 5 | 5.20 | 5.08 | 29.68 |
| | | | 3, 3 | 7.89 | 5.09 | 68.36 |
| | | | 5, 1 | 13.21 | 6.16 | 104.81 |
| | GRU | 333.79 | 1, 5 | 5.21 | 4.91 | 19.65 |
| | | | 3, 3 | 7.80 | 4.49 | 44.93 |
| | | | 5, 1 | 12.96 | 5.14 | 69.41 |
| | Stacked_Bi_GRU | 2976.06 | 1, 5 | 7.82 | 7.13 | 68.48 |
| | | | 3, 3 | 11.30 | 9.17 | 166.58 |
| | | | 5, 1 | 17.40 | 11.90 | 274.43 |
| | CNN_LSTM | 1703.61 | 1, 5 | 9.49 | 7.28 | 41.11 |
| | | | 3, 3 | 12.91 | 9.55 | 100.99 |
| | | | 5, 1 | 18.90 | 13.45 | 173.57 |
| | Deep_CNN | 143.17 | 1, 5 | 5.62 | 6.20 | 13.03 |
| | | | 3, 3 | 8.60 | 7.45 | 29.32 |
| | | | 5, 1 | 14.37 | 9.21 | 44.42 |
| | IndRNN | 142.34 | 1, 5 | 4.72 | 4.36 | 14.54 |
| | | | 3, 3 | 6.89 | 3.42 | 33.95 |
| | | | 5, 1 | 11.22 | 2.99 | 56.54 |

**Table 5** Comparison among LSTM, Bi-LSTM, GRU, Stacked_Bi_GRU, CNN_LSTM, Deep_CNN, and IndRNN in terms of params.

| Model | Total params | Fine-tuning params |
|-------|--------------|--------------------|
| LSTM | 89217 | 129 |
| Bi-LSTM | 178433 | 257 |
| GRU | 66945 | 129 |
| Stacked_Bi_GRU | 1110905 | 569 |
| CNN_LSTM | 688161 | 673 |
| Deep_CNN | 13121 | 65 |
| IndRNN | 6145 | 129 |

almost overlapped, which demonstrates the effectiveness of the fine-tuned IndRNN model in predicting the development of COVID-19 cases.

## Predictive performance analysis of the COVID-19 epidemic situations in six countries

In this section, the proposed fine-tuned IndRNN model is used to predict the COVID-19 epidemic trend in the cumulative confirmed and death cases of the China, the United States, India, Brazil, France and Russia with 5 fine-tuning data. The prediction period is one month, according to the fact that the impact of policies or events generally exhibits in the future month. And then the actual data is utilized to validate the predict result. The development tendency of the COVID-19 is analyzed, the significant phenomenon is then detected and the reasons that responsible for this is further explored.

Figure 7 shows the development tendency and prediction result of COVID-19 cumulative confirmed and death cases in China, and the general development stages of the epidemic in China are summarized in the Table S1. As can be seen from the figure, the cumulative confirmed cases in China experienced dramatic changes. It firstly starts to rise sharply in mid-January 2020, while the number of deaths continues to rise, and the epidemic has entered the outbreak stage (*The State Council Information Office of the People's Republic of China, 2020*). During this period, the Chinese government issued a timely emergency response, *e.g.*, expanding the laboratories capacities for nucleic acid detection, the construction of 'Huoshenshan', 'Leishenshan', and Fangcang shelter hospitals for accommodating more patients, medical observation was conducted on close contacts, people remained at home as much as possible, and wearing masks, avoiding gatherings, and keeping a physical distance of 1 meter when going out were strictly required (*Li et al., 2020b*; *Chinese Center for Disease Control and Prevention, 2020a*). The China CDC issued prevention guidelines for people of different ages and places to strengthen the awareness of safety (*Chinese Center for Disease Control and Prevention, 2020b*). As a result, the epidemic has been promptly controlled, and the cumulative confirmed cases in China has gradually flattened and stabilized by late April 2020. The COVID-19 deaths in China have been basically flat since the end of April 2020, and have remained at 4,636 since January 25th, 2021. It is demonstrated that the epidemic in China has been adequately controlled since the late April 2020.

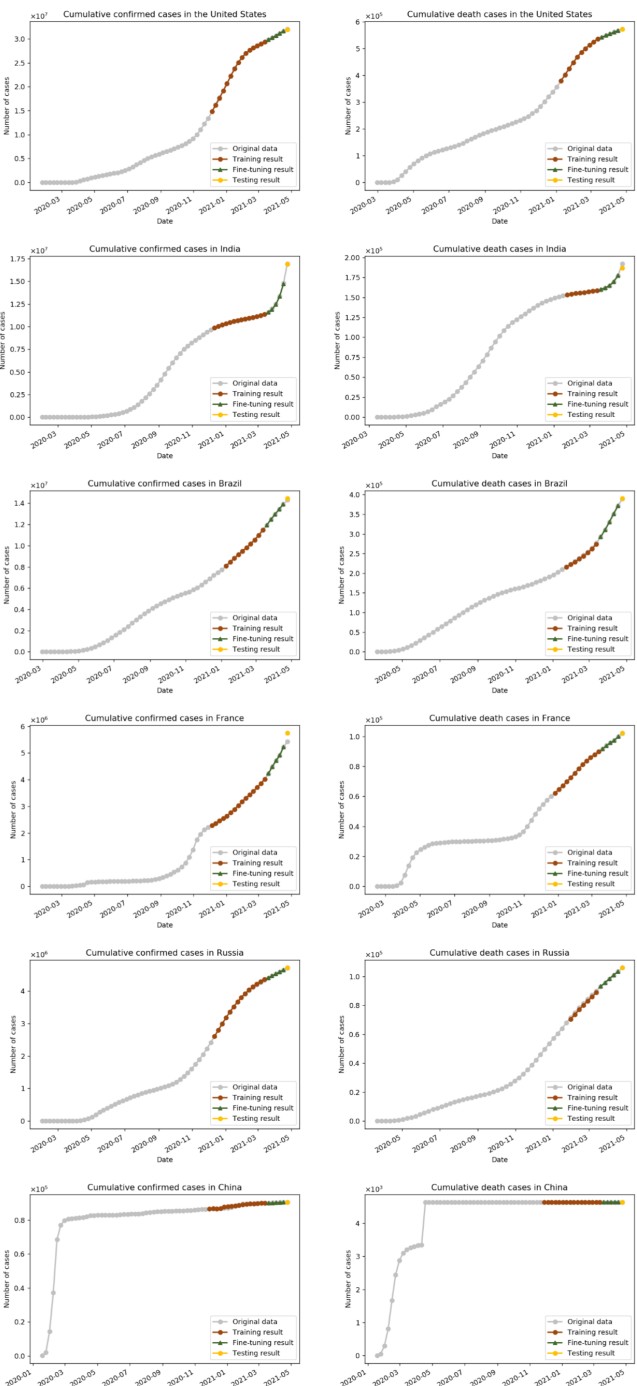

**Figure 6** The fitting performance of the fine-tuned IndRNN.

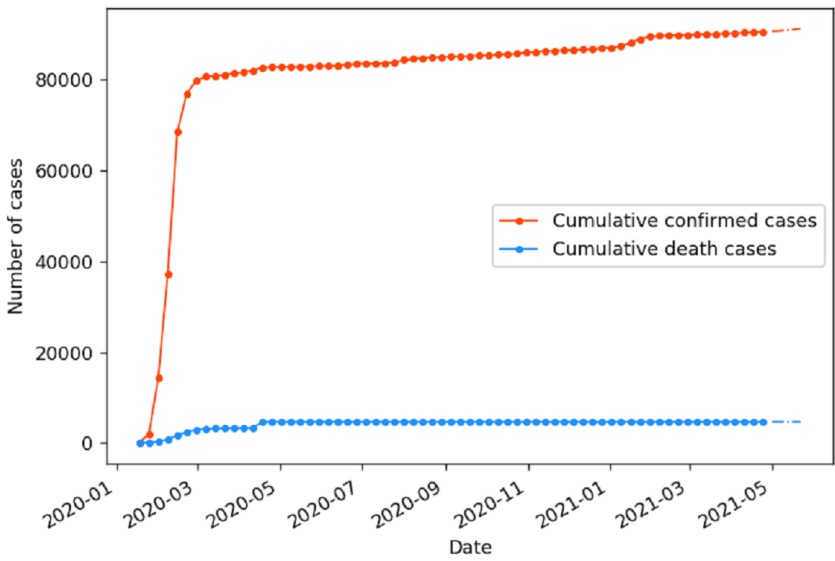

**Figure 7** **Prediction of cumulative COVID-19 cases in China by the fine-tuned IndRNN.** The solid line is the actual value and the dashed line is the predicted value.

Figure 8 illustrates the development trend of cumulative confirmed and death cases in the United States, India, Brazil, France, Russia. Different from that of China, the growth of cumulative cases of the top five countries is relatively slow before July 2020. After that, it began to increase sharply, especially for the United States and India. The rudimentary development of the COVID-19 epidemic in the United States is summarized in the Table S2. After the first COVID-19 patient in the United States was recorded on January 20th, 2020 (*Holshue et al., 2020*), the following measures were taken: non-Americans who had visited China in the past 14 days were banned from entering on February 2th, 2020 (*National Immigration Administration, 2020*), cruise ship ban, and masks were promoted by the CDC (*Adams, 2020*). Nevertheless, there were still a campaign of mass protests and campaign rallies without protective measures (*Schuchat, 2020*). As a result, the account of confirmed cases in the United States continued to increase significantly after surpassing China on March 26th, 2020. In addition, the 'opening up' measures promoted by the American government (*Good Morning America, 2020a*), and the opening of schools and holding of the rally about the presidential campaign since August 2020 (*Mansfield, Salman & Voyles Pulver, 2020*), promote the intensification of the epidemic, hence the cumulative confirmed cases in the United Stated start to rise sharply in October 2020.

The India has no obvious growth before July 2020, in which the 'city closure' measures implemented by the Indian government at the end of March have played a key role (*Xinhua, 2020a*). But some areas have been re-blocked, and the number of people suffering from the COVID-19 in India has increased extraordinary, because of an outbreak rebound caused by the increase of outdoor activities after a gradual unsealing since June 2020 (*Xinhua, 2020a*; *Associated Press, 2020*). Many massive rallies had been held in India since March

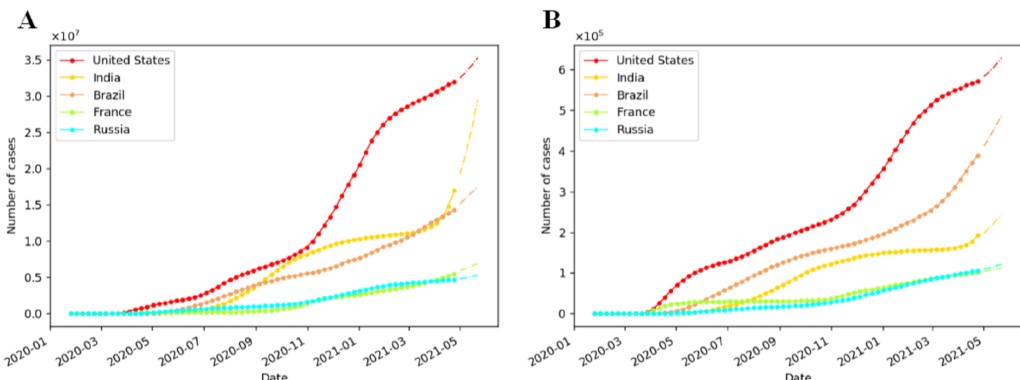

**Figure 8** **Prediction of cumulative COVID-19 cases in five countries by the fine-tuned IndRNN.** (A) Prediction of weekly cumulative confirmed cases. (B) Prediction of weekly cumulative death cases. The solid line is the actual data, and the dotted line is the forecast data.

2021, including political rallies and festival (*Bhuyan, 2021*), and the number of cumulative confirmed cases has surged.

Brazil, France, and Russia do not show a significant increase in general, but the number of cases in Brazil is higher than that in France and Russia. Brazil implemented a social distancing policy, but some people neglected to wear masks at rallies (*The paper, 2020*), and professional football matches were held in Rio, Brazil on June 18th, 2020 (*Xinhua, 2020b*), these events promoted the development of the epidemic. France implemented a lockdown on March 17th, 2020, and gradually unblocked it from May 11th, 2020. However, the amount of cumulative diagnoses in France starts to rise in October 2020, by reason of the emergence of cases in school and surges in cases in many areas, hence the French government closed cities for a second time on October 30th, 2020 (*Government of France, 2020*; *Bell & Bairin, 2020*); The Russian government did not take timely measures to prevent the European epidemic, causing a large number of imported cases (*Kanka News, 2020*); moreover, the holding of a military parade on June 24th, 2020 (*Xinhua, 2020c*), has become one of the factors driving the curve of Russia upward.

According to Figs. 7 and 8, the trend forecasts in six countries are basically in line with the curve trend. Table 6 is the testing and validation accuracy of cumulative confirmed and death COVID-19 cases in five countries by IndRNN model with fine-tuning. As indicated from the table, in most cases, the MAPE of testing and validation are less than 1.2%, and 6.2%, respectively. In the table, it generates lowest MAPE and RMSE values of 0.04% and 2.00 in testing results, and 0.05% and 1.17 in verification results, respectively, for deaths in China, which shows the effectiveness of fitting and prediction by the proposed framework.

## Verification of prediction accuracy in global diagnosis

The IndRNN model was used to predict epidemic trends by training the cumulative confirmed weekly cases for the global from February 15th, 2020 to April 24th, 2021. Figure 9 displays the prediction diagram. By May 22th, 2021, the number of global cumulative confirmed cases may surpass 171.4 million, and deaths may reach 3.5 million,

**Table 6** Testing and validation of cumulative confirmed and death COVID-19 cases in five countries by the fine-tuned IndRNN.

| Data | Country | Testing | | Validation | |
|---|---|---|---|---|---|
| | | RMSE | MAPE (%) | RMSE | MAPE (%) |
| Cumulative cases | United States | 138,071.00 | 0.43 | 1355857.54 | 3.34 |
| | India | 77,668.00 | 0.46 | 1587170.63 | 4.30 |
| | Brazil | 164,181 | 1.15 | 1027451.51 | 6.18 |
| | France | 320,132.50 | 5.89 | 719450.18 | 11.53 |
| | Russia | 19,647.00 | 0.42 | 213593.59 | 3.77 |
| | China | 17.90 | 0.02 | 187.94 | 0.17 |
| | Global | 558,483 | 0.38 | 2953263.00 | 1.60 |
| Cumulative deaths | United States | 1273.06 | 0.22 | 25260.92 | 3.72 |
| | India | 5,752.81 | 2.99 | 39954.53 | 13.92 |
| | Brazil | 1,460.31 | 0.37 | 24974.75 | 4.90 |
| | France | 300.83 | 0.29 | 3236.43 | 2.36 |
| | Russia | 194.50 | 0.18 | 3333.14 | 2.47 |
| | China | 2.00 | 0.04 | 2.14 | 0.05 |
| | Global | 5,829.50 | 0.19 | 47980.75 | 1.17 |

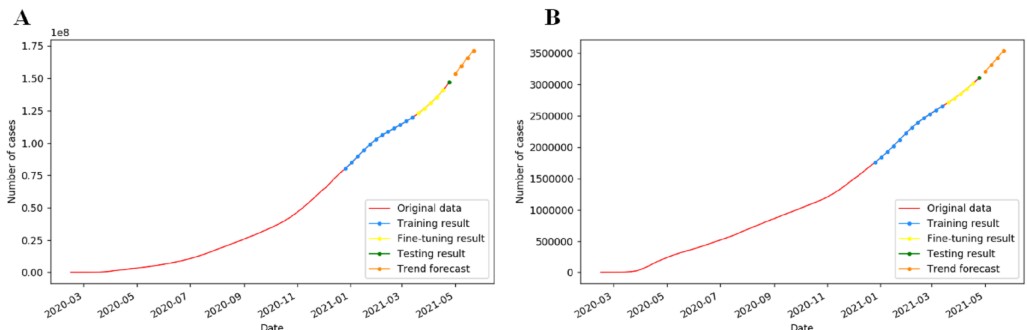

**Figure 9** **Prediction of global cumulative COVID-19 cases by the fine-tuned IndRNN.** The solid line is actual data, the dotted line represents the predicted data.

and the deviations between these results and the real data are 2.81%, and 2.30%, respectively.

## DISCUSSION

Among the seven models introduced in 'Related work', the IndRNN model with fine-tuning strategy achieved the optimal result. In different models, fine-tuning can continuously adjust the parameters in a relatively short time. Compared with the result without fine-tuning, the error is decreased by up to 20.94%. It is confirmed that the fine-tuning takes full advantage of the characteristics obtained from the pre-trained model, and this strategy plays a good role in increasing accuracy.

According to the experimental results of LSTM, Bi-LSTM, GRU, Stacked_Bi_GRU, CNN_LSTM, Deep_CNN, and IndRNN models, the IndRNN model takes the second least time, but its RMSE and MAPE are the smallest. Among the seven networks, the errors of the

stacked networks are increased compared with the simple networks, and the phenomenon of overfitting occurs, which is because the stacked networks over-interpret information, resulting in the weak generalization ability of the models; compared with CNN, the variant network (*i.e.,* LSTM, Bi-LSTM, GRU, IndRNN) of RNN has higher accuracy, due to the memory function of the internal structure of RNN series network, which is able to extract features from sequences of different moments. This ability to process time series is more consistent with the characteristics of COVID-19 epidemiological data changing over time. Among the variant networks of RNN, the accuracy of LSTM, Bi-LSTM and GRU is not much different, which is inconsistent with the result that Bi-LSTM is better than LSTM and GRU found by *Shahid, Zameer & Muneeb (2020)*. The reason is that the Bi-LSTM has insufficient learning ability on cumulative data of India with variable growth rate. By contrast, IndRNN has the best performance as the non-interference of independent neurons with each other, it is more adaptable to the changeable epidemic data, and is capable of better extract and transmit information hidden in the COVID-19 sequence data. Therefore, it can be considered that the IndRNN model can understand the regular pattern of epidemic data, has good learning capabilities, and achieves the coexistence of high precision and low time consumption.

The MAPE of the testing by IndRNN model with fine-tuning is less than 1.2%, except for 5.89% of the cumulative confirmed cases in France and 2.99% of the cumulative deaths in India, and the MAPE of the validation by this framework is less than 6.2%, except for 11.53% of the cumulative confirmed cases in France and 13.92% of the cumulative deaths in India. This is due to the sudden increase in the cases and the short duration of the growth trend, which affects the ability of model to obtain stable features and predict trend.

The more fine-tuning data is used, and the more error is reduced after fine-tuning, owing to enough fine-tuning data can make full use of the features extracted by the pre-trained model. Concretely, the reduced errors of LSTM, Bi-LSTM, GRU, Stacked_Bi_GRU, CNN_LSTM, Deep_CNN, and IndRNN models after fine-tuning are 0.27%~12.36%, 0.12%~14.29%, 0.3%~10.91%, 0.69%~8.89%, 2.21%~11.84%, −1.59%~13.79%, 0.36%~20.94%, respectively. About Deep_CNN model, there is a slight increase in the error after fine-tuning when the number of fine-tuning data is 1, and one possible reason is that too few fine-tuning parameters and data affect the effect of parameter adjustment. The degree of error reduction is affected by the ability of the model to learn features and the fine-tuned model to learn features from the pre-trained model, and the different changes in different data, as well as the amount of fine-tuning data. Among them, the fine-tuning model using 5 fine-tuning data makes full use of features obtained from pre-trained IndRNN model, and get best result.

The model speed is proportional to the number of parameters, but not to the fitting effect. The parameters of IndRNN model are the least by reason of no gate calculations in internal units in compare with LSTM, GRU, Bi-LSTM units. However, as the three gates and memory cell in the recurrent unit of the LSTM model are more than the GRU model with two gates, and Bi-LSTM model has bidirectional LSTM layer in the internal structure, hence, LSTM model has more parameters than GRU model, and the parameters of Bi-LSTM model are about twice that of the LSTM model. Stacked_Bi_GRU and CNN_LSTM have

**Table 7  Positive and negative events.**

| Type | Event |
| --- | --- |
| Positive | (1) Release of emergency response |
| | (2) Seal off the city, close entertainment venues |
| | (3) The prohibition of activities |
| | (4) Establishment of temporary hospital |
| | (5) Border inspection |
| | (6) Cruise ship ban |
| | (7) Prevention and control guidelines |
| | (8) Vigorously produce medical materials, improvement of the detection ability of COVID-19 |
| Negative | (1) Relax restrictions too early for opening |
| | (2) Policy implementation is not timely |
| | (3) Hosying of great events: demonstration, campaign, festival |
| | (4) Face masks are not mandatory |

a large number of parameters due to the stack of multiple network layers, which affects the running speed. Among the seven networks, Deep_CNN is the model with the fewest fine-tuning parameters because of a small number of dense layer parameters caused by characteristic of shared parameters, hence the fine-tuning speed is the fastest. However, the test RMSE of this model is up to 9.22% higher than that of IndRNN, which may be because the reduction of parameters affects the effect of feature extraction of sequence data.

Through the policies mentioned above, we divide these policies into active measures and inactive measures according to whether they contribute to the mitigation of the epidemic (Table 7). We conclude that, on the final results of epidemic prevention and control, the timely release of prevention and control policies by the government is the most important. Local implementation and cooperation of residents are also crucial. The COVID-19 epidemic of China gradually declines to a stable state in April 2020 as good performance in these respects. Some countries have adopted the same policy as China but the effect is not obvious, *e.g.*, the United States, India, owing to the deficiency of implementation of some regions and collaboration of residents, such as no mask, organizing of the rally. The short implementation time of blockade measures in some countries promoted the recurrence of the epidemic, for instance, India. Policy relaxation and economic restarts should be conducted after the epidemic situation is relatively stable, rather than prematurely. Large-scale activities with crowd gatherings should be cancelled as much as possible, owing to the existence of a large potential infection rate. At the resident level, some countries have promulgated corresponding measures, but citizens have not strictly implemented them. Some citizens resist the 'closing the city' measure, for they think it conflicts with personal freedom. Some people have negative emotions about the severe epidemic and determination of them to protest together is insufficient, affecting the policy response. It is essential to actively respond to the relevant prevention and control policies issued by the state, and not to violate the prohibition without authorisation. Although the

current epidemic situation in some countries is gradually stable, keeping vigilant, paying attention to personal protection, and reducing long-distance travel remain important.

To ensure that the training data is more than the fine-tuning data to get good results, if the time interval is long, it is necessary to download the continuously updated data for retraining again, and then use the subsequent data for fine-tuning, so as to achieve the goal of not needing to re-train the data in a short time and improving the accuracy.

## CONCLUSIONS

In this work, deep learning models were utilized to research the development of COVID-19 in China, the top five countries and the world. The framework based on IndRNN and fine-tuning consists of four steps: data preprocessing, pre-training the model and saving the weight, fine-tuning the weight, trend prediction and validation. The development tendency of COVID-19 was analyzed, predicted, and validated. Some conclusions are draw as follows:

(1) The validity of the proposed framework is verified by comparing it with LSTM, Bi-LSTM, GRU, Stacked_Bi_GRU, CNN_LSTM, and Deep_CNN. The result demonstrated that IndRNN model shows the best performance, and has low complexity. Compared with no-fine-tuning, the fine-tuned IndRNN model can effectively the reduce the prediction errors by up to 20.94% and the time cost. For most of the countries, the MAPE of IndRNN model with fine-tuning was less than 1.2%, and the lowest MAPE and RMSE values of 0.04% and 2.00 in testing results were gendered, for deaths in China, which indicated the effectiveness of the proposed framework.

(2) According the prediction and validation results, the MAPE of IndRNN model with fine-tuning was less than 6.2% in most cases, and this framework obtained a minimum MAPE and RMSE of 0.05% and 1.17, respectively, by using Chinese deaths. The deviations between predicted cumulative confirmed and death cases on the world in late May 2021 and the real data were 2.81%, and 2.30%, respectively, which confirmed the predictive performance of the proposed framework.

(3) Policies play an important role in the development of COVID-19. Timely and appropriate measures can greatly reduce the spread of COVID-19. Some countries have adopted the same positive policy but the effect is not ideal due to the untimely and inappropriate government policies, low implementation ability and the coordination degree of the public. The positive measures include release of emergency response, sealing off cities and closing entertainment venues, prohibiting activities, establishment of temporary hospital, border inspection, cruise ship ban, the release of prevention and control guidelines, the vigorously production of medical materials, improvement of the detection ability of COVID-19. The negative measures are relaxing restrictions too early for opening, timely policy implementation, hosting of great events, and that face masks are not mandatory. Additionally, untimely and inappropriate government policies, lax regulations, and insufficient public cooperation are the reasons for the aggravation of the epidemic situations.

## ACKNOWLEDGEMENTS

We appreciate the World Health Organization (WHO), National Health Commission of the people's Republic of China, and the Johns Hopkins University in the United States for providing us with the use some of information or data.

### Funding

This work was supported by the National Key R&D Program of China under Grant 2018YFB0505400 and the National Natural Science Foundation of China under Grant 41871325. The funders had no role in study design, data collection and analysis, decision to publish, or preparation of the manuscript.

### Grant Disclosures

The following grant information was disclosed by the authors:
The National Key R&D Program of China: 2018YFB0505400.
National Natural Science Foundation of China: 41871325.

### Competing Interests

The authors declare there are no competing interests.

### Author Contributions

- Zhonghua Hong and Ziyang Fan conceived and designed the experiments, performed the experiments, analyzed the data, performed the computation work, prepared figures and/or tables, authored or reviewed drafts of the paper, and approved the final draft.
- Xiaohua Tong conceived and designed the experiments, analyzed the data, authored or reviewed drafts of the paper, and approved the final draft.
- Ruyan Zhou performed the experiments, analyzed the data, authored or reviewed drafts of the paper, and approved the final draft.
- Haiyan Pan performed the experiments, analyzed the data, prepared figures and/or tables, authored or reviewed drafts of the paper, and approved the final draft.
- Yun Zhang, Yanling Han, Jing Wang and Shuhu Yang analyzed the data, authored or reviewed drafts of the paper, and approved the final draft.
- Hong Wu and Jiahao Li performed the experiments, analyzed the data, performed the computation work, authored or reviewed drafts of the paper, and approved the final draft.

### Data Availability

The code is available at GitHub: https://github.com/zhhongsh/COVID19-Precdiction.
The prediction by the fine-tuned IndRNN are available at: http://47.117.160.245:8088/IndRNNPredict.

## Supplemental Information

Supplemental information for this article can be found online at http://dx.doi.org/10.7717/peerj-cs.770#supplemental-information.

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

## FURTHER READING

**Abcnews. 2020c.** Mass protests could lead to another wave of coronavirus infections. *Available at* https://abcnews.go.com/-US/mass-protests-lead-wave-coronavirus-infections/story?id=70997184 (accessed on 14 August 2020).

**China Central Television. 2020.** The National Health Commission reminds again during the May Day holiday: time-sharing appointments for peak-shift trips. *Available at* http://news.cctv.com/2020/05/02/ARTIlsOonxEyXZQdSG4ioeV6200502.shtml (accessed on 14 August 2020).

**Chinanews. 2020.** From Huoshenshan, Leishenshan to the Fangcang hospital, race against the epidemic at the speed of China! (in Chinese). *Available at* http://www.chinanews.com/sh/shipin/cns/2020/02-05/news846922.shtml (accessed on 14 August 2020).

**Good Morning America. 2020b.** Trump reacts after low turnout at Tulsa rally. *Available at* https://www.goodmorningamerica.com/news/video/trump-reacts-low-turnout-tulsa-rally-71379690 (accessed on 14 August 2020).

**Hubei Provincial People 's Government. 2020a.** Announcement of Wuhan City's COVID-19 Epidemic Prevention and Control Headquarters (in Chinese). *Available at* http://www.hubei.gov.cn/zhuanti/2020/gzxxgzbd/zxtb/202001/t20200123_2014402.shtml (accessed on 14 August 2020).

**Hubei Provincial People 's Government. 2020b.** At 0:00 today, Wuhan unblocked (in Chinese). *Available at* http://www.hubei.gov.cn/2019/tpy-w/202004/t20200408_2207205.shtml (accessed on 14 August 2020).

**Human Resources and Social security Department of Hubei Province. 2020.** The decision on awarding Dingyu Zhang and Jixian Zhang for great achievements. *Available at* http://m.hbdysh.cn/article/25927 (in Chinese) (accessed on 14 August 2020).

**Jernigan DB. 2020.** Update: public health response to the coronavirus disease 2019 outbreak—United States, February 24, 2020. *Morbidity and Mortality Weekly Report* **69**:216–219 14 August 2020 DOI 10.15585/mmwr.mm6908e1.

**Pathan RK, Biswas M, Khandaker MU. 2020.** Time series prediction of covid-19 by mutation rate analysis using recurrent neural network-based LSTM model. *Chaos Solitons & Fractals* **138**:110018 DOI 10.1016/j.chaos.2020.110018.

**People.cn. 2020.** Provide strong legal protection for epidemic prevention and control (in Chinese). *Available at* http://yuqing.people.com.cn/n1/2020/0211/c429781-31580633.html (accessed on 14 August 2020).

**Xinhuanet. 2020d.** Spotlight: more U.S. states mandate wearing masks in public as COVID-19 cases continue to rise. *Available at* http://www.xinhuanet.com/english/2020-06/25/c_139166307.htm (accessed on 14 August 2020).

**Xinhuanet. 2020e.** Restart epidemic briefing. Trump calls for wearing masks (in Chinese). *Available at http://m.xinhuane-t.com/world/2020-07/23/c_1210716033.htm* (accessed on 14 August 2020).

**Xinhuanet. 2020f.** Only one week after school starts, the number of confirmed cases in U.S. universities surges. *Available at http://www.xinhuanet.com/video/2020-08/26/c_1210770179.htm* (accessed on 31 August 2020).