# Peer review of "Prediction of COVID-19 epidemic situation via fine-tuned IndRNN"

_PeerJ Computer Science, doi:10.7717/peerj-cs.770_

## Round 0.1 · original submission · Major Revisions

· Academic Editor

Major Revisions

The authors should address review comments to revise the manuscript.

Reviewer 1 ·

Basic reporting

No comment.

Experimental design

I think the details of data processing in the experimental design, including all variables from the preprocessing and coding of the original obtained data to how to train and test, should be clearer and add more details.

Validity of the findings

I suggest that the discussion of the results should be more fully compared with the previous studies. Please clarify and add.

Additional comments

The structure of this article is complete and professional, with clear diagrams and tables, and a clear sharing of materials. However, I think the reference literature, especially the collection of the literature on the epidemic, is slightly insufficient. Not only must we collect more literature about COVID-19, but even from the perspective of public health, I think it is necessary to go back to the Spanish flu... etc. The relevant literature on the spread of the epidemic is discussed in order to complete the discussion in the background.
For your research design, I think the details of data processing in the experimental design, including all variables from the preprocessing and coding of the original obtained data to how to train and test, should be clearer and add more details.
Finally, I suggest that the discussion of the results should be more fully compared with the previous studies. Please clarify and add.

Reviewer 2 ·

Basic reporting

Many flaws in the article that are needed to be addressed by authors.

Experimental design

Need to be revised in terms of phases and draw a main figure the tell the whole story.

Validity of the findings

No benchmark scenario is found, so how authors can prove that their study better than existing ones.

Additional comments

The authors presented very traditional study which I think it is more to evaluation rather than development. However, several limitations are exist:
- The title is too general and it reflects the contribution more to evaluation rather than development study ?
- The abstract is too long and need to be revised in the context of academic article.
- Authors mentioned "Therefore, it is necessary to propose a model to predict and analyze the development tendencyof COVID-19, and to summarise the meaningful and positive policies"
this is too general challenge which was handled by many studies. Furthermore, the contribution of the study is very poor. Moreover, the authors need to be specific about the contributions of the study.
- Comparison with state of the art studies also is missing.
- The related works section is missing. Thus, the study failed to fill up to highlight the gap of existing studies.

---

## Round 0.2 · accepted · Accept

· Academic Editor

Accept

The reviewer has recommended accepting the manuscript.

Reviewer 2 ·

Basic reporting

Authors have addressed well all issues and the articles in the present format can be accepted for publication.

Experimental design

Well organized research methodology

Validity of the findings

Robust validation scenario